# OpenReview forum: "MLE-bench: Evaluating Machine Learning Agents on Machine Learning Engineering"
_ICLR.cc/2025/Conference — ICLR 2025 Oral_

### Official Review · Reviewer_HGZk · 2024-10-26

**Soundness:** 3
**Presentation:** 3
**Contribution:** 3
**Rating:** 6
**Confidence:** 4

**Summary:**

This paper constructs a benchmark for evaluating the capabilities of LLMs in automated data science tasks based on Kaggle competitions. This paper presents MLE-bench, consisting of 71 competitions covering diverse data modalities and task complexities. Rule-breaking detection and plagiarism detection are provided to prevent LLMs from generating undesired behaviors. Evaluations are conducted on both closed-source and open-source LLMs.

**Strengths:**

1. The proposed benchmark is quite challenging for current LLM agents.

2. A lot of factors are considered in the benchmark, such as time constraint, computation resource, etc.

3. The empirical evaluation is comprehensive.

**Weaknesses:**

1. My major concern lies in the test splits used in this benchmark, which cannot guarantee alignment with the private leaderboard score and thus leads to unfair comparison with data scientists from Kaggle. I checked several Kaggle competitions used in this benchmark and found that they supported “late submission” to provide the leaderboard score. Why MLE-bench does not choose to fully leverage this feature? Could you check whether the test splits used in MLE-bench align with the realistic leaderboard via this feature? If not, I think MLE-bench can only provide comparison among LLM agents rather than human data scientists from Kaggle.

2. As discussed in Line 501-504, there is an obvious discrepancy on available machine learning techniques between past Kagglers and modern LLMs. I suspect that the capabilities of LLM agents for winning a medal in MLE-bench also correlates to the year of the Kaggle competition. As such, the evaluation metric solely relying on whether an LLM agent can win a medal may lead to biased conclusion. Could you present complete results to show the effect of the competition starting year for the performance?

3. Also, could you provide empirical analyses on the effect of the competition complexity (as labeled high/medium/low) for the agent performance?

4. Could you present some successful cases and failed cases of o1-preview in MLE-bench?  The trajectories shown in Figure 2 are not complete enough to derive insightful findings.

5. How does plagiarism detection work? If the current competition is A, the plagiarism is detected in terms of merely notebooks in A or all the notebooks in competitions from MLE-bench?

6. As discussed in Line 505-509, I also think the current benchmark is too heavy for potential future research purposes. Maybe a light-weight version of MLE-bench can be considered as future work.

**Questions:**

See weaknesses

---

> ### Author Response · Authors · 2024-11-20
>
> Thank you for taking the time to carefully review our paper! We’ll address your comments below:
>
> > My major concern lies in the test splits used in this benchmark, which cannot guarantee alignment with the private leaderboard score and thus leads to unfair comparison with data scientists from Kaggle. I checked several Kaggle competitions used in this benchmark and found that they supported “late submission” to provide the leaderboard score. Why MLE-bench does not choose to fully leverage this feature? Could you check whether the test splits used in MLE-bench align with the realistic leaderboard via this feature? If not, I think MLE-bench can only provide comparison among LLM agents rather than human data scientists from Kaggle.
>
> The “late submission” feature unfortunately places too much reliance on Kaggle infrastructure, and e.g. has rate limits and daily submission limits which are not ideal for the longevity of the benchmark.
>
> As discussed in Section 6 on Limitations, we acknowledge that there is uncertainty in the alignment between the private leaderboard score and MLE-bench grading, but we feel that the effect size will be small given our care in producing similar test splits (See Table 7).
>
> We agree that an experiment to compare agent solutions graded via Late Submission VS MLE-bench would help resolve this question, but it is quite a difficult experiment to run (we can’t grade the same submission.csv for each setting since the test sets differ, so we would have to manually curate solutions for each), and unfortunately we will not do doing this now.
>
> > I suspect that the capabilities of LLM agents for winning a medal in MLE-bench also correlates to the year of the Kaggle competition. As such, the evaluation metric solely relying on whether an LLM agent can win a medal may lead to biased conclusion. Could you present complete results to show the effect of the competition starting year for the performance?
>
> That’s a sensible hypothesis! We have added this plot as Figure 9 in the appendix, which finds no strong correlation between agent performance and competition year – though we notably find that agents fail to score models on the most recent comps that occurred in 2023-2024 (which are also most difficult by today’s standards).
>
> > Also, could you provide empirical analyses on the effect of the competition complexity (as labeled high/medium/low) for the agent performance?
>
> Yes, we have added this result in Table 9 in the Appendix, showing the breakdown of results by competition complexity. We find that the complexity labels correspond well to the agent’s performance, as expected.
>
> > Could you present some successful cases and failed cases of o1-preview in MLE-bench? The trajectories shown in Figure 2 are not complete enough to derive insightful findings.
>
> Thanks for asking! We’re currently looking into releasing our agent transcripts alongside our code release but we’ll need an additional step of approval from our organization for this. We agree this would be a good thing to share and hope to share an update soon.
>
> > How does plagiarism detection work? If the current competition is A, the plagiarism is detected in terms of merely notebooks in A or all the notebooks in competitions from MLE-bench?
>
> Good question! For a given competition A, we only check for plagiarism in the notebooks associated with competition A, not all the competitions. We’ve clarified Section A.4 in the paper to describe this more clearly.
>
> > As discussed in Line 505-509, I also think the current benchmark is too heavy for potential future research purposes. Maybe a light-weight version of MLE-bench can be considered as future work.
>
> Thank you for highlighting this! We’ve given this careful thought and decided that a good option is to encourage users to make use of the `Low` complexity split of our dataset for lightweight evaluation (this split contains 22 competitions instead of 71, and skews toward datasets and hardware requirements that are more lightweight). We’ve updated Section 2 of our paper to mention this option, and added Table 9 in our paper to include metrics for each of the complexity splits for comparison. We will also make this option clear in our public messaging around the benchmark later, via channels separate to the paper. We hope that this option will improve the accessibility of our benchmark!
>
> ---
>
> Once again, thank you for your thoughts and valuable feedback! We hope we have addressed most of your concerns. Please consider raising your review score if you feel this process has improved the quality of our paper.

---

> > ### Comment · Reviewer_HGZk · 2024-11-21
> >
> > Thanks for your detailed response. I think most of concerns are fully addressed.
> >
> > Could you explain more about why an (even several competitions in MLE-Bench are enough) experiment for comparison between the estimated medal gain rate via your test splits and late submission is hard to perform currently? I cannot figure out where the difficulty comes from:
> > > We agree that an experiment to compare agent solutions graded via Late Submission VS MLE-bench would help resolve this question, but it is quite a difficult experiment to run (we can’t grade the same submission.csv for each setting since the test sets differ, so we would have to manually curate solutions for each), and unfortunately we will not do doing this now.
> >
> > Why the test sets differ? I think the test sets are fixed for a Kaggle competition.

---

> ### Author Response · Authors · 2024-11-21
>
> Sure, let me try to be clearer! There are two slightly different grading scenarios:
> 1. **Kaggle Grading (KG)** The original online Kaggle competition, using the original test set (also used for the Late Submission)
> 2. **MLE-bench Grading (MG)** Our test sets constructed for MLE-bench, carved out from the original train set. (Described in Section 2.1 and Appendix A.7)
>
> We are interested to know if there are any differences between the KG and MG grading, such that scores on MLE-bench (measured via MG) may not be comparable to leaderboard scores on Kaggle (KG).
>
> To study the differences, ideally we would like to take a submission $S$, grade it using both KG and MG to obtain $f_\text{KG}(S)$ and $f_\text{MG}(S)$, then measure the difference $f_\text{KG}(S) - f_\text{MG}(S)$. The complication is that since KG and MG have different test sets (KG uses the original hidden test set, MG uses our custom made test set), we cannot use a single submission.csv $S$, since the predictions for each test set would be different.
>
> Instead, we can obtain a program $P$ which, given a train and test set $D$, trains a model and produces a submission.csv for each setting. We can use this program $P$ to obtain $S_\text{KG} = P(D_\text{KG})$ and $S_\text{MG} = P(D_\text{MG})$. Finally, we would measure the difference $f_\text{KG}(S_\text{KG}) - f_\text{MG}(S\text{MG})$.
>
> To complete this experiment, we'll need to
> 1) obtain a suitable set of programs $P$ for each competition (we'll have to manually source these either from existing Kaggle solutions or find suitable programs from our existing agent attempts),
> 2) run them to train models and predict solutions, and
> 3) grade the solutions on KG and MG.
>
> Overall, the experiment is not conceptually difficult but we estimate that this could take a few days to execute well and report results on. It is not an unreasonable amount of time, but it is difficult for our team given our current availabilities.
>
> Finally, we'd like to point out that **we have already done a variant of this experiment**, by comparing the scores of the Sample Submissions on KG and MG (this allowed us to skip steps 1 and 2 above) as a quality check when implementing our graders. This makes us less worried about this being a serious problem. Please see our reply to Reviewer gB21 [here](https://openreview.net/forum?id=6s5uXNWGIh&noteId=EbnQhqNJ1P) about Sample Submissions for more details.
>
> (If there is a simpler version of this experiment that you had in mind, we'd be excited to hear it!)

---

> ### Comment · Reviewer_HGZk · 2024-11-22
>
> The three steps basically align with my expectations except that I thought that the checkpoints of the trained models were saved so that such checks can be easily performed.
>
> I also checked the mentioned variant experiment but I didn’t find any quantitative results.
>
> I still think this is a major concern since public leaderboard and private leaderboard often present misalignment in Kaggle competitions. Even minor differences in the evaluation metrics can lead to different medals. **The evaluation here should be rigorous enough to claim whether an agent can achieve a medal.** Thus, from my perspective, an experiment (a small subset of competitions is enough) to demonstrate this point is necessary.

---

### Official Review · Reviewer_gB21 · 2024-10-29

**Soundness:** 4
**Presentation:** 4
**Contribution:** 4
**Rating:** 10
**Confidence:** 4

**Summary:**

The authors present MLE-bench, a benchmark designed to evaluate AI agents’ capabilities in machine learning engineering. This benchmark is constructed from publicly available Kaggle challenges, with AI agent performance assessed based on the percentage achieving medal-level scores comparable to real human submissions. The paper releases the benchmark’s data and code, includes three open-source agents as baselines, and evaluates state-of-the-art foundational models using the AIDE agent.

**Strengths:**

1. Given recent advancements in AI agents for coding, engineering, and scientific discovery, the proposed benchmark serves as a highly valuable testbed for evaluating foundational models and agents in real-world machine learning engineering contexts.
1. The paper is clearly written and easy to follow, providing necessary details on both the dataset and technical implementation.
1. The study includes solid empirical evaluations across different agents, foundational models, scaling factors, and contamination issues, offering valuable insights into AI agents’ current capabilities and limitations in the field.
1. The authors provide a thorough discussion of the benchmark’s limitations and ethical considerations.

**Weaknesses:**

### Accessibility of the Benchmark

As discussed by the authors in Section 6 L505, the benchmark is costly to run. Based on my estimates, using the authors’ cost descriptions and current rates on OpenAI and Lambda Cloud, a single full evaluation with AIDE and o1-preview would require approximately 4,000 USD (around 2,600 USD for API queries and 1,300 USD for an A10 GPU server). Considering the costs for development and extensive experiments would be substantially higher than a single evaluation, this benchmark may be inaccessible to small to medium-scale academic labs.

The authors may consider providing a lighter dataset split to improve accessibility, similar to the approach in SWE-Bench.

### Evaluation Reliability and Contamination

One particular challenge in benchmarks for AI agents is establishing reliable detection for potential cheating behaviors (e.g., hacking the evaluation function or accessing private test data). I appreciate the authors for addressing this with a tool designed to flag such behaviors. However, Table 6 in Appendix A.3 indicates a high false positive rate, which may hinder practical reliability. The rate is sufficiently high that manually checking all flagged submissions would be demanding.

Additionally, while the authors provide a thorough discussion and empirical analysis of contamination issues, this remains a critical limitation for this and other similar benchmarks. For instance, Figure 5 shows GPT-4o familiarity scores above 0.4 across all problems. Does this suggest that these problems are included in the model’s training set? Also, the conclusions drawn from the correlation between familiarity and performance could be significantly impacted by confounders, such as problem difficulty. Furthermore, while the obfuscated dataset and plagiarism detection tools are commendable efforts, foundational models could still potentially recognize rephrased questions and apply high-level strategies from their memories, making such behaviors difficult to detect.

More discussions on those concerns could be helpful. However, it is worth noting that these issues reflect broader challenges in the field, and it would be unreasonable to expect any single paper to fully resolve them. The paper has made valuable contributions with its detailed discussions and insightful empirical results on these challenges.

### Comparison Between Human Medal Results

As noted by the authors in Section 6 (L497), the train-test splits used in MLE-bench differ from the original splits in Kaggle competitions. The authors state that they “ensure that the distributions… are similar by checking that the example submission scores similarly on both sets” (L156). However, it is unclear how these example submissions were created. If the same model training pipeline were applied to both the original training set and the modified training set (a subset of the original), one would typically expect lower performance on the latter due to reduced training data. Could the authors clarify the configuration of the example submission, and specifically, what comparisons were made and under which settings?

### Additional Questions

I have some further questions, outlined below, that may also be worth addressing in the paper.

**Questions:**

1. Would the authors consider discussing related works on AI agents, AutoML, and automated scientific discovery in the Related Work section? These areas seem relevant to the benchmark’s objectives.
1. Regarding the difficulty estimation in L145, how reliable is the human estimation process? Could the authors provide additional details on the setup and methodology for these annotations?
1. In L300, the authors note that “agents would execute commands that overload the machine’s disk or RAM, resulting in their process getting killed and their run finishing early.” Do the tested agents incorporate any error-handling or reflection mechanisms for such situations?
1. Are the three results in Table 3 statistically different from one another? It would be challenging to interpret the higher performance of the extra GPU setting if the second GPU was not utilized, which might suggest that differences could arise merely from noise.
1. In Figure 3, it might be useful to further scale o1-preview, as the curve does not yet appear to have plateaued.

---

> ### Author Response · Authors · 2024-11-20
>
> First of all, thank you very much for your thoughtful review; we’re thrilled that you think this is a valuable evaluation!
>
> > The authors may consider providing a lighter dataset split to improve accessibility, similar to the approach in SWE-Bench.
>
> Thanks for the suggestion! We’ve given this careful thought and decided that a good option is to encourage users to make use of the `Low` complexity split of our dataset for lightweight evaluation (this split contains 22 competitions instead of 71, and skews toward datasets and hardware requirements that are more lightweight). We’ve updated Section 2 of our paper to mention this option, and added Table 9 in our paper to include metrics for each of the complexity splits for comparison. We will also make this option clear in our public messaging around the benchmark later, via channels separate to the paper.
>
> > Table 6 in Appendix A.3 indicates a high false positive rate, which may hinder practical reliability. The rate is sufficiently high that manually checking all flagged submissions would be demanding.
>
> We agree that the classifier is imperfect, and we were careful to separate this from the core of the benchmark so it can be easily swapped out with another tool. We hope that tooling will improve in the near future as models and scaffolds improve.
>
> > Figure 5 shows GPT-4o familiarity scores above 0.4 across all problems. Does this suggest that these problems are included in the model’s training set?
>
> Good question! Not necessarily; the familiarity scores are computed from the base model’s log probabilities across all the tokens in the competition pages. Since the pages are written in English, it is expected for the token probabilities to have a baseline level of familiarity simply from common sentence structure and phrases.
>
> > the conclusions drawn from the correlation between familiarity and performance could be significantly impacted by confounders, such as problem difficulty.
>
> It is true that we cannot rule out such confounders, though each familiarity level has a mix of competition complexities, so the confounding effects of difficulty should be averaged out.
>
> > If the same model training pipeline were applied to both the original training set and the modified training set (a subset of the original), one would typically expect lower performance on the latter due to reduced training data.
>
> Good point! We’ve been careful in constructing our splits (as detailed in A7) to ensure that the training set is not significantly reduced in size. We agree that future experiments like the one you suggested would provide helpful evidence, though we will not pursue it now.
>
> > Could the authors clarify the configuration of the example submission, and specifically, what comparisons were made and under which settings?
>
> Sure! The sample submission is competition-specific, but corresponds to a dummy answer (e.g. predicting “dog” for every input in a classification problem). We do the following:
>
> 1. Use the same logic for constructing the sample submission as is specified in the competition description to make an equivalent sample submission for our test split.
> 2. Grade our sample submission locally on our benchmark.
> 3. Compare the score our sample submission achieves on our benchmark to the score the online sample submission achieves on the Kaggle leaderboard.
>
> Because the sample submission construction logic is identical between the local and online split, verifying that the scores are in line with each other acts as one sanity check on our grading implementation and splits. (Our codebase is now available as Supplementary Material, and contains all the sample submissions used.)
>
> > Regarding the difficulty estimation in L145, how reliable is the human estimation process? Could the authors provide additional details on the setup and methodology for these annotations?
>
> The complexity was annotated by an engineer on our team using the definition in Section 2.1, and reviewed by at least one other engineer. We've also added a breakdown of agent results by complexity in Table 9, showing that the complexity labels correspond well to the agent’s performance.
>
> > “agents would execute commands that overload the machine’s disk or RAM, resulting in their process getting killed and their run finishing early.” Do the tested agents incorporate any error-handling or reflection mechanisms for such situations?
>
> Great question! For disk space, there is no error-handling mechanism — if the machine runs out of space, the agent will crash. For RAM, the behavior depends on the agent scaffold. For example, OpenHands executes actions in a separate process, and if the process exceeds the available RAM, it occasionally throws a Python MemoryError that the main process can catch and recover from.
>
> > In Figure 3, it might be useful to further scale o1-preview
>
> We agree that scaling o1-preview further would likely improve performance, but we won't run further experiments here due to the associated costs.

---

> > ### Comment · Reviewer_gB21 · 2024-11-21
> >
> > Thank you for addressing my questions. I believe your responses thoroughly address the issues raised. I will maintain my score and recommend acceptance of the paper.

---

### Official Review · Reviewer_uxvJ · 2024-11-01

**Soundness:** 3
**Presentation:** 3
**Contribution:** 3
**Rating:** 8
**Confidence:** 4

**Summary:**

This paper introduces a new benchmark, “MLE-Bench”, with the goal of assessing AI agent’s abilities at machine learning engineering (MLE) tasks. The benchmark consists of 71 hand-selected Kaggle competitions from domains including image classification (38%), text classification (14%), and tabular tasks (13%) at a variety of difficulty levels. For each task, the agent is provided with a description of the task and a dataset. The agent is prompted to solve the task by writing code which is executed and evaluated analogously to the process of evaluating human Kaggle contestants’ submissions. Each agent is given up to 24 hours to iteratively improve its solution before evaluation.

The authors use MLE-Bench to evaluate several combinations of agent scaffolds and base language models,

- AIDE (with o1-preview, gpt-4o, llama-3.1-405b, claude-3-5-sonnet),
- ResearchAgent from MLAgentBench (with gpt-4o),
- CodeActAgent from OpenHands (with gpt-4o),

and compare the performance of these agents with a baseline derived from the performance of human Kaggle contestants. The evaluation metric is the percentage of tasks in which the agent would have received a Bronze, Silver, or Gold medal, had it actually participated in the respective Kaggle competition.

The authors show that both the choice of agent scaffold as well as the choice of language model has a significant effect on performance, and show that the best agent receives a medal in 17.3% of tasks.

The paper also includes experiments on the effect of hardware provided to the agents (0-2 GPUs) and the amount of time available to agents (up to 100 hours per task). Further, the authors evaluate whether the agent’s potential familiarity with a task (i.e. inclusion in its training data) affects performance. Finally, they analyze the agent’s code outputs for potential rule violations and plagiarism.

**Strengths:**

The experimental results presented in the paper are interesting and provide a clearer picture on current AI agent’s abilities for MLE. The benchmark will be a useful contribution to future work on agent scaffolds and LLMs, as well as for evaluating current systems from AI safety and preparedness perspectives.

Some strengths worth highlighting:

1. The proposed benchmark is comprehensive, and covers many parts of ML engineering (preprocessing, model training,...) and task domains (image, text, tabular,..).
2. The main experiments are rigorous and demonstrate the usefulness of the proposed benchmark.
3. The evaluation protocol seems generally sensible and the reported metrics are useful for understanding the results, although some improvements could be made (see below).

**Weaknesses:**

Please see the following suggestions that, in my opinion, would significantly improve this paper. Given some updates to the paper and clarifications to the questions in the next section, I would be happy to increase the score.

1. The authors acknowledge that their benchmark is very resource-intensive to run, requiring 1704 GPU-hours and >100M LLM tokens for a single seed (see Sec 6./Accessibility). Given that their main results used 16 and 36 seeds, this is clearly inaccessible to a large fraction of the research community. The paper would be improved if the authors could provide a lighter version of the benchmark (e.g. a subset of 5-10 tasks that reflect the main challenges of MLE) along with the metrics on this subset. This would not require running any additional experiments.
2. The aggregated results (across tasks, seeds, and time steps) provided in the paper are useful, but do not provide a full picture of the remaining open challenges in MLE and why the agents achieve good/bad levels of performance. It would be great if the authors could include more detailed and also qualitative results. In particular:
    1. A clearer analysis of which kinds of tasks agents perform well or badly on (e.g. split scores by complexity level and by task domain).
    2. The paper mentions raw per-task scores (Sec 2.2), although these do not seem to be included in the paper.
    3. The paper mentions that the authors analyzed agent transcripts/logs. It would be useful if these transcripts were provided (not necessarily in the paper, but in an external source)

**Questions:**

1. In Section 3.3, the authors show that the number of GPUs (0-2) provided to the agent does not significantly affect performance. This is a surprising result as, in contrast, I would expect this to make a substantial difference for human data scientists/MLEs. Could you share any insights regarding this? How often/rarely are agents using a GPU if it is provided? Does the majority of medals come from tasks where no GPU is necessary?
2. The paper uses Bronze/Silver/Gold medals to evaluate performance, which quantizes the leaderboard ranking into top-40%/top-20%/top-10% buckets (with varying thresholds as described in Sec. 2.2). Why did you choose this evaluation over using the leaderboard ranking directly (normalized into [0, 1])? That would likely give more fine-grained performance information.
3. In Figure 5, you use the “score normalized between the sample submission score and the gold medal score for that competition”. Why did you use this new metric instead of either the fraction of medals across seeds (as in other experiments) or the leaderboard ranking?
4. Depending on the agent, in 20% or more cases the agent is unable to make any valid task submission. What are generally the reasons for this? Including these reasons in the paper could support future work on improving these agents.

---

> ### Author Response · Authors · 2024-11-20
>
> Thank you for taking the time to carefully review our paper! We'll address your comments below:
>
> > The paper would be improved if the authors could provide a lighter version of the benchmark along with the metrics on this subset
>
> Great suggestion! We’ve given this careful thought and decided that a good option is to encourage users to make use of the `Low` complexity split of our dataset for lightweight evaluation (this split contains 22 competitions instead of 71, and skews toward datasets and hardware requirements that are more lightweight). We’ve updated Section 2 of our paper to mention this option, and added Table 9 in our paper to include metrics for each of the complexity splits for comparison. We will also make this option clear in our public messaging around the benchmark later, via channels separate to the paper. We hope that this option will improve the accessibility of our benchmark!
>
> > A clearer analysis of which kinds of tasks agents perform well or badly on (e.g. split scores by complexity level and by task domain).
>
> We’ve added Table 9 and Table 10 in the Appendix to provide this breakdown of scores by complexity and task domain, as requested.
>
> > The paper mentions raw per-task scores (Sec 2.2), although these do not seem to be included in the paper.
>
> We’ve uploaded our codebase as a zip file in the supplementary material, and included the full grading reports for all our experiments in the `runs/` folder. This will also be included in our public Github codebase release.
>
> > The paper mentions that the authors analyzed agent transcripts/logs. It would be useful if these transcripts were provided
>
> Thanks for asking! We’re looking into this, we’ll need an additional step of approval from our organization to release agent transcripts but we agree this would be a good thing to share and hope to share an update soon.
>
> > Re: GPU performance: Could you share any insights regarding this? How often/rarely are agents using a GPU if it is provided? Does the majority of medals come from tasks where no GPU is necessary?
>
> We agree that this is surprising. From our experience, we see that agents often write programs that use the GPU if it is available, though this occurs as part of a boilerplate step (e.g. `device = torch.device("cuda" if torch.cuda.is_available() else "cpu")`) and we don’t see agents focusing on GPU performance. It appears that the majority of medals come from tasks not requiring GPUs, given that the no-GPU setup has a similar number of medals. One hypothesis is that tasks not requiring large GPUs tend to have smaller datasets, which are simpler to work with and easier to iterate on which is why agents naturally do better on these.
>
> > Re: Leaderboard metrics: Why did you choose this evaluation over using the leaderboard ranking directly (normalized into [0, 1])?
>
> Good question! Although it’s natural to think so, medals are not simply a quantization of leaderboard percentile. Kaggle has their own carefully calibrated system with specific rules to decide how and when medals are awarded, such that the value of each medal type reflects the same quality of achievement regardless of e.g. how many participants you competed against (see https://www.kaggle.com/progression). We choose to rely on Kaggle’s definition which has been more robustly tested, and also believe that medals makes for salient thresholds, i.e. the Kaggle community is accustomed to comparing how many medals different users have, and Kaggle Grandmasters are defined according to medal count.
>
> > Re: Figure 5: Why did you use this new metric instead of either the fraction of medals across seeds (as in other experiments) or the leaderboard ranking?
>
> We ran this experiment with GPT-4o AIDE, which gets a medal only 8.7% of the time. If we used medals on the y-axis, each point would be quantized down and we were worried that the performance would be too weak to get a meaningful signal here. (Note that although this normalized score has more resolution, the floor is not consistently defined across competitions, so we still prefer our medals-based metric as our main metric.)
>
> > Depending on the agent, in 20% or more cases the agent is unable to make any valid task submission. What are generally the reasons for this?
>
> To make a submission, the agent has to produce a submission file at `/home/submission/submission.csv`. This fails to count as a valid submission if the agent does not produce such a file, OR the file does not contain data in the correct format. We’ve seen failures of all kinds, e.g. where the agent fails to do the work necessary to produce a prediction; the agent forgets to write its predictions to a file; the agent writes to the wrong path; the submission does not have the correct number of predictions, etc.
>
> Once again, thank you for your thoughts and valuable feedback! We hope we have addressed most of your concerns. Please consider raising your review score if you feel this process has improved the quality of our paper.

---

> > ### Comment · Reviewer_uxvJ · 2024-11-23
> >
> > Thank you for the detailed answers and clarifications to my questions, and for providing the supplementary material! I'm increasing my rating.

---

### Official Review · Reviewer_fmGi · 2024-11-04

**Soundness:** 3
**Presentation:** 3
**Contribution:** 3
**Rating:** 8
**Confidence:** 3

**Summary:**

This paper introduces MLE-bench, which is a benchmark consisting of 71 Kaggle competitions for measuring how well AI agents perform at machine learning engineering. The benchmark includes human baselines using the publicly-available Kaggle leaderboards. The paper also includes benchmarking with various scaffolds and foundation models, as well as some analysis of possible issues like contamination from pre-training.

**Strengths:**

Originality. To my knowledge, this benchmark is the first that is designed to evaluate ML engineering capabilities of AI agents.

Quality. The paper has done a good-faith effort to benchmark using relevant models and mitigate potential issues (e.g., contamination and plagiarism). I liked the transparent discussion of limitations and potential issues.

Clarity. I found the paper generally easy to follow.

Significance. The introduction of this benchmark is quite timely given the interest in developing high-quality software engineering agents.

**Weaknesses:**

1. There seems to be an issue with Figure 2. I can only see a small snippet of the figure.

2. I am concerned about the accessibility of this benchmark. As stated in section 6, it is a resource-intensive benchmark to run. If I understand the cost breakdown correctly, a single seed costs over $2500 to run (for the current prices of o1-preview). This is simply not feasible to university labs. I would suggest the authors provide two versions of the benchmark: one that is more accessible and one that is less accessible.

3. Given that the main contribution of this work is the benchmark, I think some of the experiments could be pushed to the appendix, whereas more details about the benchmark could be in the main body. For example, I’d like to see a more clear setup and rules section to make using the benchmark as easy as possible.

4. Given that this is a datasets and benchmarks submission, I wish the anonymized codebase was made available during submission. As a result, I am also reducing my confidence score.

5. There was a lot of discussion about splitting the competitions based on complexity, but I don’t see any presentation of the agent scores as a function of complexity. It feels strange to have this decomposition without using it in the later analysis.

6. Some of the selection criteria is clear (e.g., completed competition), but others are more qualitative (e.g., well-specified description), so it would be nice to see something a bit more detailed and systematic for those.

7. Some of the choices are not super clear. For example, what does “where sensible, we maintain the train/test split ratio” mean? (L157-158). Similarly, why was the headline metric chosen that way? Is this standard for Kaggle competitions?

8. I would have liked to see examples of the generated code, potentially with an additional quality analysis. This analysis need not be extensive, even conducting the analysis on a randomly-selected output for one task would be interesting.

**Questions:**

1. Could the authors please clarify how the complexity for each competition was derived? A common way to do this would be calculate Cohen’s kappa on independently labeled by annotators. Is there a computed agreement score? (L145-149). I feel like this part could be more clear and principled.

2. How were the 7 development competitions chosen? (L150-152).

3. Are the restrictions in Section 3 a part of the benchmark? For example, the time limit of 24 hours? (L243).

4. Is the plagiarism checker provided as a part of the benchmark for free? (L229-233).

---

> ### Author Response · Authors · 2024-11-20
>
> Thank you for taking the time to carefully review our paper! We'll address your comments below:
>
> > There seems to be an issue with Figure 2. I can only see a small snippet of the figure.
>
> Thank you for catching this! We replicated the issue in Safari and uploaded a fixed version.
>
> >  I would suggest the authors provide two versions of the benchmark: one that is more accessible and one that is less accessible.
>
> Thanks for the suggestion! We’ve given this careful thought and decided that a good option is to encourage users to make use of the `Low` complexity split of our dataset for lightweight evaluation (22 competitions instead of 71, and skews toward datasets and hardware requirements that are more lightweight). We’ve updated Section 2 of our paper to highlight this option, and added Table 9 in our paper to include metrics for each of the complexity splits for comparison. We'll also make this option clear in our public messaging around the benchmark later.
>
> > I’d like to see a more clear setup and rules section to make using the benchmark as easy as possible.
>
> Thank you for the feedback! Could you clarify which aspects of the setup and rules were unclear? We’d be happy to address them in the next revision.
>
> > I wish the anonymized codebase was made available during submission.
>
> We have now uploaded a zip of the codebase as Supplementary Materials, and we will release this as a public Github repository as well.
>
> > I don’t see any presentation of the agent scores as a function of complexity.
>
> Thanks for the suggestion! We’ve added Table 9 which breaks down agent performance by complexity level.
>
> > Some of the selection criteria is clear (e.g., completed competition), but others are more qualitative (e.g., well-specified description), so it would be nice to see something a bit more detailed and systematic for those.
>
> Thank you for raising this point! We’ve updated Appendix A1 to clarify what we meant by “well-specified”: The description is detailed and thorough without any major ambiguities about how to implement the competition that might only be resolved in the Discussion tab or external materials. In practice, there were very few borderline cases and it was often clear whether the competition was well-specified or not.
>
> > what does “where sensible, we maintain the train/test split ratio” mean? (L157-158).
>
> We’ve updated Section 2.1 to be more explicit about our process for determining train/test split sizes. It was challenging to find a single hard-and-fast rule suitable for all competitions. We therefore followed the rule “take 10% of the original training set for the new test split” except for where it didn’t make sense.
>
> For example, the “New York City Taxi Fare Prediction” competition has 5.42M train samples and 9k test samples. Here, using the 10% rule would give our new test set two orders of magnitude more samples than the original, so we opted to instead maintain a similar train/test ratio to the original.
>
> > Similarly, why was the headline metric chosen that way? Is this standard for Kaggle competitions?
>
> Yes, medals are the standard metric used in Kaggle competitions. Basing our headline metric on medals has some advantages: Kaggle has a carefully calibrated system to decide how and when medals are awarded, such that the value of each medal type reflects the same quality of achievement regardless of e.g. how many participants you competed against. For more details, see https://www.kaggle.com/progression. Furthermore, medals have an intuitive interpretation to the general public.
>
> > I would have liked to see examples of the generated code, potentially with an additional quality analysis.
>
> Thank you for the suggestion! We’re currently seeking permission to share more examples and will provide an update once we receive a response.
>
> > Could the authors please clarify how the complexity for each competition was derived?
>
> The complexity was annotated by an engineer on our team using the definition in Section 2.1, and reviewed by at least one other engineer.
>
> > How were the 7 development competitions chosen? (L150-152).
>
> The development competitions were selected for their small dataset sizes, which ensures they’re quick to download and fast to iterate on during development.
>
> > Are the restrictions in Section 3 a part of the benchmark? For example, the time limit of 24 hours? (L243).
>
> Good question! The details of our particular setup outlined in Section 3 are not requirements of the benchmark because we don’t want the benchmark to be hardware or resource specific.
>
> > Is the plagiarism checker provided as a part of the benchmark for free? (L229-233).
>
> Yes, the plagiarism checker, Dolos, is open source, though we don't redistribute it. Once installed you can call it from our code.
>
> ---
>
> Once again, thank you for your valuable feedback! We hope we have addressed most of your concerns. Please consider raising your review score if you feel this process has improved the quality of our paper.

---

> > ### Comment · Reviewer_fmGi · 2024-11-24
> >
> > Thank you for your detailed response! I feel that most of my questions have been answered, but I am still on the fence about a few points regarding the rigor (e.g., the grading point raised by Reviewer HGZk). However, given that most of my questions have been sufficiently answered and I think this is overall a nice paper, I will raise my score accordingly.

---

### Meta-Review · Area_Chair_z5Eu · 2024-12-19

**Metareview:**

This paper proposes a new benchmark that evaluates LLMs on the ability to solve ML engineering tasks taken from Kaggle. Reviewers overall liked this paper and found the experiments to be well-done and comprehensive. I agree with the reviewers that this benchmark could be useful, especially as ML engineering tasks would be a common application.

Strengths of the paper:
1. Comprehensive and useful tasks
2. Experiments are well-done.

Weakness of the paper:
1. Full benchmark is resource-intensive
2. Some concerns raised by reviewer HGZk regarding mis-alignment with the private leaderboard. I think this is indeed a concern but I think this doesn't affect comparing one LLM agent with another which will be the most common scenario. Authors mention that given time they can run the aligned experiments so I do encourage them to report those numbers and comment on any discrepancies.

Overall, I recommend acceptance.

**Additional Comments On Reviewer Discussion:**

Reviewers raised concerns

1. Experiments being resource-intensive
2. Lack of analysis and code being unavailable
3. Concerns regarding misalignment with the private leaderboard leading to challenges of comparison LLMs with data scientists (described above)
4. Concerns that LLM agents today have access to more ML techniques than human scientists in the past and thus have an "easier" job.

These are important concerns. Authors responded to these by suggesting using a subset of the benchmark that needs fewer resources (1),
providing the code for (2), providing additional analysis for (3), and running an experiment that shows that LLMs won medals in the past over years except in recent time for (4) (Figure 9). The last concern is something that bothers me since failure to win medals in the last 2 years could potentially hint at LLMs doing well in the past due to more information about past competitions available on the internet. E.g., questions on StackOverflow that might be similar to what was asked on Kaggle. These could be hard to detect.

For reasons (3) and (4), I think it is a stretch to compare the performance of LLM agents and human data scientists who participated in that competition. That said, I believe the dominant use case will be comparing LLM agents with LLM agents and so I think this benchmark will still end up being useful and, therefore, I recommend acceptance. I would urge authors to clarify that scores cannot be easily compared with human judgment. Also, it might help to have a "correlation score" accompanying Figure 9 that is easier to parse.

---

### Decision · Program_Chairs · 2025-01-22

Accept (Oral)